# ECG Synthesis via Diffusion-Based State Space Augmented Transformer

**DOI:** 10.3390/s23198328

**Published:** 2023-10-09

**Authors:** Md Haider Zama, Friedhelm Schwenker

**Affiliations:** 1Department of Computer Engineering, Jamia Millia Islamia, New Delhi 110025, India; haider1272002@gmail.com; 2Institute of Neural Information Processing, Ulm University, 89081 Ulm, Germany

**Keywords:** electrocardiography, generative models, diffusion models, signal processing, time series, ECG synthesis

## Abstract

Cardiovascular diseases (CVDs) are a major global health concern, causing significant morbidity and mortality. AI’s integration with healthcare offers promising solutions, with data-driven techniques, including ECG analysis, emerging as powerful tools. However, privacy concerns pose a major barrier to distributing healthcare data for addressing data-driven CVD classification. To address confidentiality issues related to sensitive health data distribution, we propose leveraging artificially synthesized data generation. Our contribution introduces a novel diffusion-based model coupled with a State Space Augmented Transformer. This synthesizes conditional 12-lead electrocardiograms based on the 12 multilabeled heart rhythm classes of the PTB-XL dataset, with each lead depicting the heart’s electrical activity from different viewpoints. Recent advances establish diffusion models as groundbreaking generative tools, while the State Space Augmented Transformer captures long-term dependencies in time series data. The quality of generated samples was assessed using metrics like Dynamic Time Warping (DTW) and Maximum Mean Discrepancy (MMD). To evaluate authenticity, we assessed the similarity of performance of a pre-trained classifier on both generated and real ECG samples.

## 1. Introduction

Amidst the landscape of global health challenges, cardiovascular diseases (CVDs) stand as a monumental concern, contributing significantly to morbidity and mortality rates. As the leading cause of death worldwide, their impact on public health and healthcare systems cannot be overstated. In recent years, the synergy between healthcare and artificial intelligence (AI) has yielded promising avenues for addressing these challenges. Data-driven AI techniques, in particular, have emerged as powerful tools in various medical domains, including the analysis of electrocardiograms (ECGs)—a cornerstone in diagnosing and managing CVDs.

Electrocardiography, commonly known as ECG, is a non-invasive method for monitoring the electrical activity of the heart over time. It provides critical insights into cardiac health and aids in the early detection, diagnosis, and monitoring of a wide range of heart-related conditions. ECG analysis involves the systematic interpretation of the heart’s electrical signals, serving as a vital tool in the diagnosis of cardiac conditions. These ECG waveforms encompass distinct components like the P waves, QRS complexes, and T waves, each of which signifies specific phases of the cardiac cycle. The analysis encompasses the evaluation of factors such as heart rate, irregular rhythms (such as arrhythmias), deviations in the heart’s electrical axis, and changes in the ST segment and T wave patterns to gauge the overall health of the heart muscle. Its clinical applications are extensive and encompass the diagnosis of arrhythmias, the identification of myocardial ischemia and infarction, the continuous monitoring of pre-existing heart conditions, the assessment of treatment effects, and the evaluation of cardiac risk before surgical procedures.

Various AI methods are applied in the analysis and classification of ECG data, with a prominent focus on convolutional architectures, as highlighted in recent reviews [1,2]. Extensive research has demonstrated the effectiveness of modern convolutional architectures like ResNet and Inception in comparison to other methods, particularly on datasets like PTB-XL [3,4,5]. Additionally, innovative approaches have been proposed in the field, such as using large recurrent neural networks with convolutional feature extraction [6]. Furthermore, recent advancements involve the utilization of Structured State Space Models (SSMs) for classification, showing superior performance compared to both convolutional and recurrent network-based methods [7]. However, the effective utilization of ECG data for disease detection and diagnosis is not without its challenges.

One major impediment in data-driven CVDs is the difficulty in the distribution of healthcare data, primarily due to privacy concerns. Health data, including ECG recordings, contain sensitive information that demands strict protection to ensure patient privacy and adhere to data regulations. Consequently, the sharing and distribution of ECG datasets, crucial for training AI models, can be hampered by stringent privacy regulations. This limitation poses a significant obstacle to the development of accurate and robust ECG classification models.

The existing data protection techniques, while essential, often fall short of providing a comprehensive solution as even intricate technical strategies, such as the implementation of standalone federated learning [8], cannot unilaterally guarantee comprehensive privacy safeguards as training samples from trained models can be reconstructed, as illustrated by model inversion attacks [9]. This is where the importance of artificially synthesizing ECGs comes into play using generative machine learning models, such as Generative Adversarial Networks (GANs) [10]. GANs have demonstrated remarkable capabilities in generating synthetic data that closely resemble real-world distributions. In the context of ECGs, GANs offer the potential to synthesize samples while excluding personal information pertaining to patients, enabling researchers to develop and refine AI models without compromising patient confidentiality. Several pioneering works have explored the application of GANs for ECG synthesis [11,12,13,14,15,16,17,18,19,20,21,22,23].

However, GANs are not devoid of limitations. Challenges such as mode collapse, training instability, and the generation of clinically plausible ECG signals remain areas of concern. This has led to the emergence of diffusion models as an intriguing alternative. Diffusion models [24] offer a unique approach to data generation by iteratively ’diffusing’ a simple distribution into the desired data distribution. The advantages of diffusion models include stability during training, controllable data generation, and the ability to capture long-range dependencies in the data.

Several recent works have demonstrated the efficacy of diffusion models in diverse applications, including image synthesis [25] and video synthesis [26]. Their potential to address some of the limitations of GANs makes them a promising technique for enhancing ECG synthesis. The major contributions of this study are as follows:Introduced a diffusion model for generating concise (10 s) 12-lead ECGs, incorporating a State Space Augmented Transformer [27] as a pivotal internal component.Enabled conditional generation of ECG samples, for different heart rhythms in a multilabel setting.Evaluated the quality of generated samples using metrics such as Dynamic Time Warping (DTW) and Maximum Mean Discrepancy (MMD). Additionally, to ensure authenticity, we evaluated how similar the performance of a pre-trained classifier is when applied to both generated and real ECG samples.

## 2. Related Work

The domain of deep generative modeling applied to time series data is an emerging and dynamic subfield in the realm of machine learning. This trajectory is substantially guided by the persistent advancements achieved in the domain of generative models, particularly prominent in the domain of visual imagery. Amidst a spectrum of methodologies, GANs (Generative Adversarial Networks) [10] have captured considerable attention and gained prominence as the favored technique combined with LSTMs in [11,12,13], with the attention technique in [14], with transformers in [15], or with ordinary differential equations in [16]. Some of the recent advancements in GAN-based ECG generation proposed in [17] include a two-stage generator and dual discriminators, where the generator utilizes Gaussian noise to produce ECG representations while considering heart diseases as a conditioning factor, and in [18] the authors introduced automated solutions to integrate pre-existing shape knowledge of ECGs into the generation process. Some other GAN-based approaches are [19,20,21,22,23]. In addition to GANs, alternative architectures are also proposed for ECG generation like variational autoencoders [28]. Recently, diffusion models have risen as a notably potent category of deep generative models suited for the objective of ECG synthesis exhibiting benefits beyond GANs; these models offer improved training stability and excel in generating higher-quality ECG samples [29,30,31,32].

## 3. Materials and Methods

### 3.1. Dataset

The conducted experiments were carried out using the PTB-XL dataset [3,4,5], a publicly accessible collection of electrocardiogram (ECG) data. This dataset encompasses a total of 21,837 records obtained from 18,885 distinct patients, each of whom possesses clinical 12-lead ECG data. These records consist of concise 12-lead ECG signals, each spanning a duration of 10 s and sampled at a rate of 100 Hz.

To facilitate the conditional generation of ECG samples, we employed 12 distinct heart rhythms in a multilabel setting as class conditionals. This set of labels allowed us to effectively guide the generative process.

In order to create a robust and comprehensive evaluation setup, we partitioned the dataset into three subsets: the train-set containing 17,441 samples, the test-set encompassing 2203 samples, and the validation-set comprising 2193 samples. This partitioning strategy was devised based on individual patient identifiers (person-ids), ensuring that records associated with the same person-id did not appear across different subsets. By adopting this approach, we aimed to guarantee a stringent assessment of the model’s ability to generalize its learnings. Moreover, this technique helped eliminate any potential biases that could arise due to data overlap between the distinct sets. Overall, our approach to data splitting was integral to maintaining the integrity of the evaluation process.

### 3.2. Diffusion Models

Diffusion models [24] are a category of generative models involving two processes: a forward process and a reverse process. Throughout the forward process, Gaussian noise is incrementally infused into the input sample over a sequence of T steps, following a variance schedule β (constant or learnable). This progressive introduction of noise continues until the input distribution converges to a standard Gaussian distribution N(0, 1). In contrast, during the reverse diffusion process, a neural network parameterized as θ is trained with the purpose of noise reduction from the sample. The forward process can be expressed as
(1)$$q\left(x_{1},\ldots ,x_{T}|x_{0}\right)=\prod_{t=1}^{T}q\left(x_{t}|x_{t}-1\right)$$
where $q\left(x_{t}|x_{t}-1\right)=N\left(x_{t},\sqrt{1-\beta_{t}}x_{t}-1,\beta_{t}I\right)$. Also. using the reparameterization trick, $x_{t}$ can be expressed as $x_{t}=\sqrt{\bar{\alpha_{t}}}x_{0}+\left(1-\bar{\alpha_{t}}\right)\epsilon$ for $\epsilon ∼N\left(0,1\right),\alpha_{t}=1-\beta_{t}$ and $\bar{\alpha_{t}}=\prod_{i=1}^{T}\alpha_{i}$. The reverse process learns to undo this noise-inducing procedure to recover $x_{0}$ from $x_{t}$, initiating with the pure Gaussian noise $x_{T}∼N\left(0,1\right)$. The reverse process can be expressed as
(2)$$p_{\theta }\left(x_{0},\ldots ,x_{T}-1|x_{T}\right)=\prod_{t=1}^{T}p_{\theta }\left(x_{t}-1|x_{t}\right)$$
where $p_{\theta }\left(x_{t}-1|x_{t}\right)=N\left(x_{t}-1,\mu_{\theta }\left(x_{t},t\right),\Sigma_{\theta }\left(x_{t},t\right)\right)$. The loss function for the reverse process is obtained by computing the variational lower bound:(3)$$E\left[-\log p_{\theta }\left(x_{0}\right)\right]\leq E_{q}\left[-\log\frac{p_{\theta }\left(x_{0}:T\right)}{q(x_{1}:T|x_{0})}\right]=E_{q}\left[-\log p\left(x_{T}\right)-\sum_{t\geq 1}\log\frac{p_{\theta }\left(x_{t}-1|x_{t}\right)}{q(x_{t}|x_{t}-1)}\right]=L$$

It was shown in [33] that the loss function can be further simplified as
(4)$$L_{simple}=E_{t},x0,\epsilon \left[||\epsilon -\epsilon_{\theta }\left(x_{t},t\right)|{|}^{2}\right]$$
where $\epsilon_{\theta }\left(x_{t},t\right)$ is predicted by the neural network and ϵ is the actual noise added to the sample $x_{t}$. Class-specific diffusion models can be implemented by conditioning the reverse process on a desired set of labels, denoted as ‘*c*’. In other words, this involves utilizing $\epsilon_{\theta }=\epsilon_{\theta }\left(x_{t},t,c\right).$

### 3.3. Structured State Space Models

At its core, Structured State Space Models (SSSMs) rely on a linear state space transition equation that establishes a link between a one-dimensional input sequence u(t) and a one-dimensional output sequence y(t). This connection is mediated by an N-dimensional hidden state x(t),
(5)$$x^{{}^{\prime }}\left(t\right)=Ax\left(t\right)+Bu\left(t\right),\;y\left(t\right)=Cx\left(t\right).$$

Here, A,B,C are the transformation matrices. It was shown in [34] that the typical utilization of randomly initialized parameters *A*, *B*, *C*, and *D* proves inadequate in effectively representing long-range dependencies. Hence, a category of matrices known as HiPPOs (high-order polynomial projection operators) was introduced to serve as the initialization for *A*. These HiPPO matrices are crafted with the specific objective of enabling parameter *A* to retain a memory of the input history u(t) until time *t*, thereby enhancing the ability of the state x(t) at time t to capture this historical context. In real-world applications, we commonly handle discrete sequences with some sequence length *L*. Hence, the discretized form of Equation (Equation 5) can be expressed as
(6)$$x_{k}=\bar{A}x_{k-1}+\bar{B}u_{k},y=\bar{C}x_{k},$$
where $\bar{A}={\left(I-\delta /2\cdot A\right)}^{-1}\left(I+\delta /2\cdot A\right),\;\bar{B}={\left(I-\delta /2\cdot A\right)}^{-1}\delta B,\;\bar{C}=C.$ And δ>0 is the small step size. After expanding the above recurrent representation, we obtain
(7)$$y_{k}=\bar{CA^{K}B}u_{0}+\ldots +\bar{CAB}u_{k-1}+\bar{CB}u_{k}$$

The convolutional representation is
(8)$$\begin{array}{c} y=\bar{K}*u,\\ \bar{K}\in R^{L}=(\bar{CB},\bar{CAB},\ldots ,\bar{CA^{L-1}B}). \end{array}$$

In [35], a Structured State Space Sequence model (S4) was proposed to efficiently compute Equation (Equation 8), where *A* and *B* are initialized as
(9)$$\begin{array}{c} A=A^{\left(d_{s}\right)}-PP^{T},\;B_{i}={\left(2i+1\right)}^{\frac{1}{2}},\\ \mathrm{where}\;P_{i}={\left(i+1/2\right)}^{\frac{1}{2}},\\ A_{ij}^{d_{s}}=-\left\{\begin{array}{cc} {\left(i+\frac{1}{2}\right)}^{\frac{1}{2}}{\left(j+\frac{1}{2}\right)}^{\frac{1}{2}}, & \mathrm{if}\;i>j\\ \frac{1}{2}, & \mathrm{if}\;i=j\\ -{\left(i+\frac{1}{2}\right)}^{\frac{1}{2}}{\left(j+\frac{1}{2}\right)}^{\frac{1}{2}}, & \mathrm{if}\;i<j \end{array}\right. \end{array}$$

Following that, the convolution kernel *K* described in Equation (Equation 8) can be efficiently computed with a time and space complexity of O(L). Therefore, when given an input *u*, the S4 output y=K∗u can also be computed efficiently.

### 3.4. State Space Augmented Transformer

In [27], the authors introduced SPADE (State sPace AugmenteD transformEr), a multilayer transformer model designed to capture intricate global and local information. The architecture of SPADE is illustrated in Figure 1 (left). This model utilizes a hierarchical design. At its foundational global layer, SPADE employs an SSM (State Space Model) to grasp global dependencies. As the SSM primarily offers broad global insights, the subsequent local layers empower the model to manage more intricate and nuanced local dependencies. In essence, the SSM introduces a robust structural bias that enriches global information within the inputs. For the local layer instantiation, the authors replace the full attention found in conventional transformer layers with effective pre-existing local attention techniques. Within the global layer (as shown in Figure 1, right), when presented with the input *X*, the resulting output *Y* is obtained as follows:(10)$$\begin{array}{c} X_{local}=\mathrm{Local}\left(\mathrm{LN}\left(X\right)\right),\\ X_{global}=\mathrm{SSM}\left(\mathrm{LN}\left(X\right)\right),\\ X_{a}=W\left[\mathrm{LN}\left(X_{local}\right),\mathrm{LN}\left(X_{global}\right)\right]+X,\\ Y=\mathrm{FFN}\left(\mathrm{LN}\left(X_{a}\right)\right)+X_{a}. \end{array}$$

In the above Equation (Equation 10), LN(.) represents layer normalization [36], while FFN(·) signifies a two-layer feed-forward neural network. The SSM used is S4.

It was shown in [27] that SPADE turns out to be efficient and effective in various natural language processing tasks. It outperforms existing methods in the long-range arena benchmark [37]. It also exhibits significantly improved speed and attains superior performance over the vanilla transformer [38] in autoregressive language modeling.

### 3.5. Proposed Approach: DSAT-ECG

The proposed model architecture DSAT (Diffusion State Space Augmented Transformer)-ECG takes inspiration from the SSSD-ECG [29] architecture, which is based on DiffWave, i.e., a diffusion-based model for audio synthesis [39]. In this improved design, the S4 [35] has been substituted with SPADE [27] layers. These SPADE layers now function as diffusion layers within every residual block, seamlessly integrating the diffusion embedding. Furthermore, the conventional second S4 layer has been replaced by an additional SPADE layer, which is seamlessly integrated after the addition assignment, along with conditional information. The model architecture is depicted in Figure 2.

During the training phase, the input samples ‘$x_{0}$’ are noised as a part of the forward diffusion process (see Section 3.2) at different diffusion time steps *‘t’* sampled uniformly from 1 to *T* transforming the samples to ‘$x_{t}$’. Next, the model receives ‘$x_{t}$’, ‘*t*’, and ‘*c*’ as input and outputs ‘$\epsilon_{0}$’, where ‘*c*’ is the class conditional; in this work, it is a multilabeled vector of length 12 representing 46 unique heart rhythm labels and ‘$\epsilon_{0}$’ is the predicted noise in the samples. For the model to converge, gradient descent is applied to MSE regression loss between the actual noise and the predicted noise. It is mathematically represented as
(11)$$\begin{array}{c} \nabla_{\theta }||\epsilon -\epsilon_{\theta }\left(\sqrt{\bar{\alpha_{t}}}x_{0}-\sqrt{1-\bar{\alpha_{t}}}\epsilon ,t,c\right)|{|}^{2} \end{array}$$

During the inference phase, $x_{T}$ is sampled as a pure Gaussian noise, i.e., $x_{T}∼N\left(0,1\right)$, and as a part of the reverse diffusion process (see Section 3.2) noise is iteratively removed from it, starting from diffusion time step T to 0 and transforming it to $x_{0}$. It is mathematically represented as
(12)$$\begin{array}{c} x_{t-1}=\frac{1}{\sqrt{\alpha_{t}}}\left(x_{t}-\frac{1-\alpha_{t}}{\sqrt{1-\bar{\alpha_{t}}}}\epsilon_{\theta }\left(x_{t},t,c\right)\right)+\sigma_{t}z \end{array}$$
where $\sigma_{t}$ is the fixed variance schedule and z∼N(0,1) *if t > 1*, *else z* is set to 0.

Following [21,29] we also focused on training generative models to synthesize a set of eight leads. This set includes the six precordial leads as well as two limb leads (namely, leads I and aVF in this work). Following this, we employ sampling from the two limb leads, I and aVF, to reconstruct the remaining four leads using Einthoven’s law and Goldberger’s equation as
(13)$$\begin{array}{c} III=II-I,\\ aVL=(I-III)/2,\\ aVF=(II+III)/2,\\ -aVR=(I+II)/2. \end{array}$$

## 4. Results

In this section, we describe the process through which we conducted a comprehensive assessment of the proposed model’s quality and authenticity, drawing inspiration from the evaluation approach used in [32]. The evaluation involved a direct comparison between the proposed model and several other existing models that were proposed in [29], namely SSSD-ECG, WaveGAN, and P2PGAN. We sought to offer a comprehensive perspective on how our model showcases advancements over existing methodologies, highlighting its promising implications, particularly in the realm of ECG synthesis. Figure 3 shows a 12-lead ECG sample generated by different models.

### 4.1. Training Settings

The implementation of all models was carried out utilizing the PyTorch Library, and the training process took place on Kaggle’s P100 GPU featuring 16 GB of RAM. For both the diffusion-based models, namely DSAT-ECG (ours) and SSSD-ECG, the configuration included four stacked residual layers each. These layers consisted of 256 residual channels and 256 skip channels, incorporated with a three-level diffusion embedding across dimensions of 128, 256, and 256. Both models employed a linear schedule involving 200 diffusion time steps, and the noise scheduler beta was adjusted from 0.0001 to 0.02. For optimization, the Adam optimizer was chosen with a learning rate of $2\times 10^{-4}$. The GAN-based model closely followed the configuration outlined in [29]. This encompassed a generator model size of 50, utilizing five deconvolutional blocks, and having 1000 latent dimensions. The discriminator, on the other hand, featured a model size of 50, incorporating six convolutional blocks. The optimization process employed the Adam optimizer with a learning rate of $1\times 10^{-4}$.

### 4.2. Quality Evaluation

While evaluating the quality of the ECG samples, we essentially measure how much the generated synthetic ECG samples resemble real ECG samples, encompassing visual traits and morphological attributes. To quantitatively measure the quality of these generated ECG samples, we make use of two distinct distance functions: Dynamic Time Warping (DTW) [40] and Maximum Mean Discrepancy (MMD) [41]. DTW is a method utilized to determine the similarity between two time series sequences. It finds the optimal alignment between two sequences while considering temporal distortions. It is mathematically expressed as
(14)$$D_{i,j}=f\left(x_{i},y_{j}\right)+\min\left\{D_{i,j-1},D_{i-1,j},D_{i-1,j-1}\right\}$$

The iteration is conducted for values of *i* ranging from 1 to *N* and values of *j* ranging from 1 to *M*, where *N* and *M* represent the lengths of the series *x* and *y*, respectively. Typically, the function $f(x_{i},y_{j})$ is defined as the square of the difference between $x_{i}$ and $y_{j}$, that is, ${\left(x_{i}-y_{j}\right)}^{2}$.

MMD is another technique employed to measure the dissimilarity between two sets of data, which could be probability distributions or samples. It aims to distinguish differences in distribution characteristics and can effectively detect variations in data points. It is mathematically expressed as
(15)$$\begin{array}{cc} MMD(X,Y) & =\frac{1}{n(n-1)}\sum_{i=1}^{n}\sum_{j\neq i}^{n}k\left(x_{i},x_{j}\right)+\frac{1}{m(m-1)}\sum_{i=1}^{m}\sum_{j\neq i}^{m}k\left(y_{i},y_{j}\right)-\frac{2}{nm}\sum_{i=1}^{n}\sum_{j=1}^{m}k\left(x_{i},y_{j}\right), \end{array}$$

In the context of the provided equation, the symbols hold the following interpretations: *n* represents the number of samples in the set *X*, while *m* corresponds to the number of samples in the set *Y*. The variables $x_{i}$ and $y_{i}$ denote individual samples taken from sets *X* and *Y*, respectively. Furthermore, the expression k(x,y) signifies the Gaussian radial basis function (RBF) kernel, specifically defined as $k\left(x,y\right)=\exp\left(-\frac{{∥x-y∥}^{2}}{2\sigma^{2}}\right)$, where σ stands for the kernel bandwidth parameter.

The outcomes of both distance metrics consistently demonstrate that the ECG samples produced by the proposed DSAT-ECG model are substantially more similar to the actual samples in terms of their quality (see Table 1 below).

### 4.3. Authenticity Evaluation

While evaluating the authenticity of the ECG samples, we assess the similarity in performance of a pre-trained classifier, which has been trained on actual ECG data when applied to both synthetic test samples and real ECG test samples. To accurately gauge the authenticity of these generated ECG samples, we employ two quantitative metrics: multilabel accuracy and multilabel AUROC (Area Under the Receiver Operating Characteristic Curve).

Multilabel accuracy is calculated as the ratio of the number of correctly predicted labels to the total number of labels, expressed as
(16)$$\mathrm{Multilabel}\;\mathrm{Accuracy}=\frac{\mathrm{Number}\;\mathrm{of}\;\mathrm{Correctly}\;\mathrm{Predicted}\;\mathrm{Labels}}{\mathrm{Total}\;\mathrm{Number}\;\mathrm{of}\;\mathrm{Labels}}$$

In multilabel AUROC, for each label, the ROC curve is created by plotting the true-positive rate *TPR* against the false-positive rate *FPR* at different classification thresholds. The multilabel AUROC is the average of the individual AUROC values for each label, expressed as
(17)$$\mathrm{Multilabel}\;\mathrm{AUROC}=\frac{1}{\mathrm{Number}\;\mathrm{of}\;\mathrm{Labels}}\left(\sum_{i=1}^{\mathrm{Number}\;\mathrm{of}\;\mathrm{Labels}}{\mathrm{AUROC}}_{i}\right)$$
where ${\mathrm{AUROC}}_{i}$ is the AUROC value for the *i*th label. The AUROC is calculated using the integral under the ROC curve.

For the classification task, we adopt the ResNet-1D model proposed in [42], which represents a 1D variant of the ResNet architecture. We trained the ResNet-1D model on the train-set of the PTB-XL dataset (see Section 3.1) and evaluated the classifier on the real test set and on the test sets generated by the models. The result of the classifier on different samples is shown in Table 2 below. The test shows that the classifier’s performance on samples generated by DSAT-ECG is most closest to its performance on real ECG samples compared to other models, affirming the fidelity of the DSAT-ECG model in synthesizing authentic ECG characteristics.

## 5. Discussion

The results obtained from the evaluation of our DSAT-ECG model reveal compelling insights into the efficacy and potential impact of our approach. The performance of our model, both in terms of quality and authenticity assessment, indicates notable progress in the field of ECG data synthesis.

Our quality assessment, employing the metrics Dynamic Time Warping (DTW) and Maximum Mean Discrepancy (MMD), illuminates DSAT-ECG’s ability to emulate intricate ECG patterns. The notably lower DTW and MMD scores, in contrast to competing models, highlight our model’s capability to capture nuanced temporal dynamics and the ability to preserve essential statistical properties inherent in real ECG data.

The assessment of authenticity, a key determinant in the data synthesis landscape, produced notable results for DSAT-ECG. The attainment of higher accuracy and AUROC metrics than the competing models signifies our model’s proficiency in generating data that resonate with the authenticity of genuine ECG data. This alignment is further substantiated by the observation that classifier performance on our model’s generated samples closely resembles that of real samples. These outcomes collectively reinforce the genuine nature of the synthesized data and accentuate the model’s suitability for applications requiring reliable and authentic data representation.

The progress in ECG data synthesis holds significance for both research and practical uses. The idea of improving data quality opens doors for better medical research, more accurate diagnostics, and improved clinical applications. This aligns with the need to handle private health data carefully. This is where creating artificial data becomes important, offering a way to balance data usefulness with privacy concerns.

For future research, we will focus on shifting from fixed to trainable variances for enhanced adaptability in capturing data intricacies, exploring alternative schedules beyond linear trajectories to optimize model convergence patterns, Furthermore, we plan to expand the loss function beyond MSE for better results and we will also try other variants of diffusion, like latent diffusion.

## 6. Conclusions

In conclusion, our study introduces DSAT-ECG, a State Space Augmented Transformer model trained using the diffusion technique. This approach addresses ECG data synthesis across a set of 12 distinct heart rhythms in a multilabel context, signifying the synthesis of ECG on a particular condition. The performance of the generated samples excelled across qualitative and authenticity assessments, outperforming the competitor models SSSD-ECG, WaveGAN, and P2PGAN employed for evaluation. Nevertheless, it is worth acknowledging that the DSAT-ECG samples do not completely substitute real samples for classifier training. This difference presents an opportunity for future advancements, demonstrating the potential for progress in the field of ECG synthesis in the near future. While the superiority of the proposed diffusion model over GAN-based models is evident, it is important to note that the computational time required for both training and inference is considerably longer. This observation serves as motivation for us to explore and develop faster diffusion-based models, such as the latent diffusion model, for ECG synthesis in our future work.

## Figures and Tables

**Figure 1 sensors-23-08328-f001:**
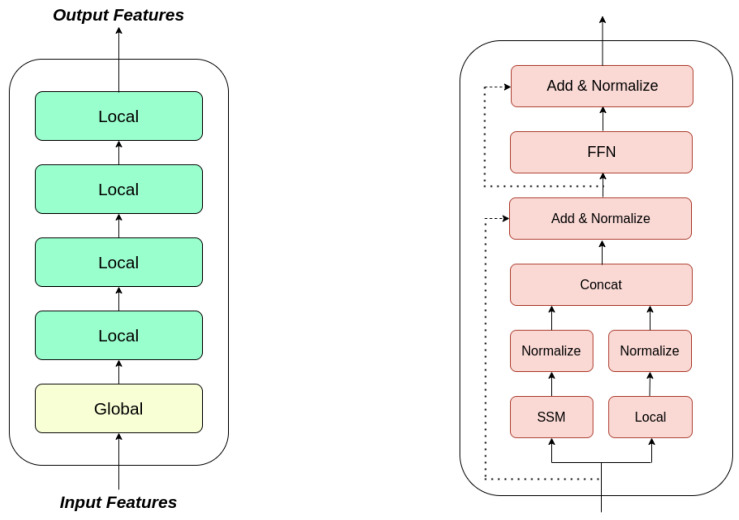
Illustration of a 4-layered SPADE: on the left: an outline of the model; on the right: intricacies of the global layer.

**Figure 2 sensors-23-08328-f002:**
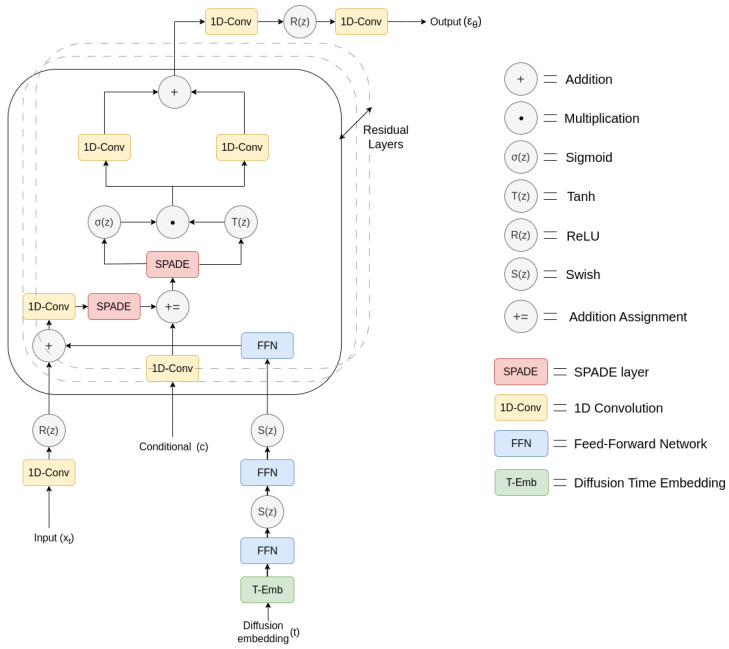
Proposed DSAT-ECG model architecture.

**Figure 3 sensors-23-08328-f003:**
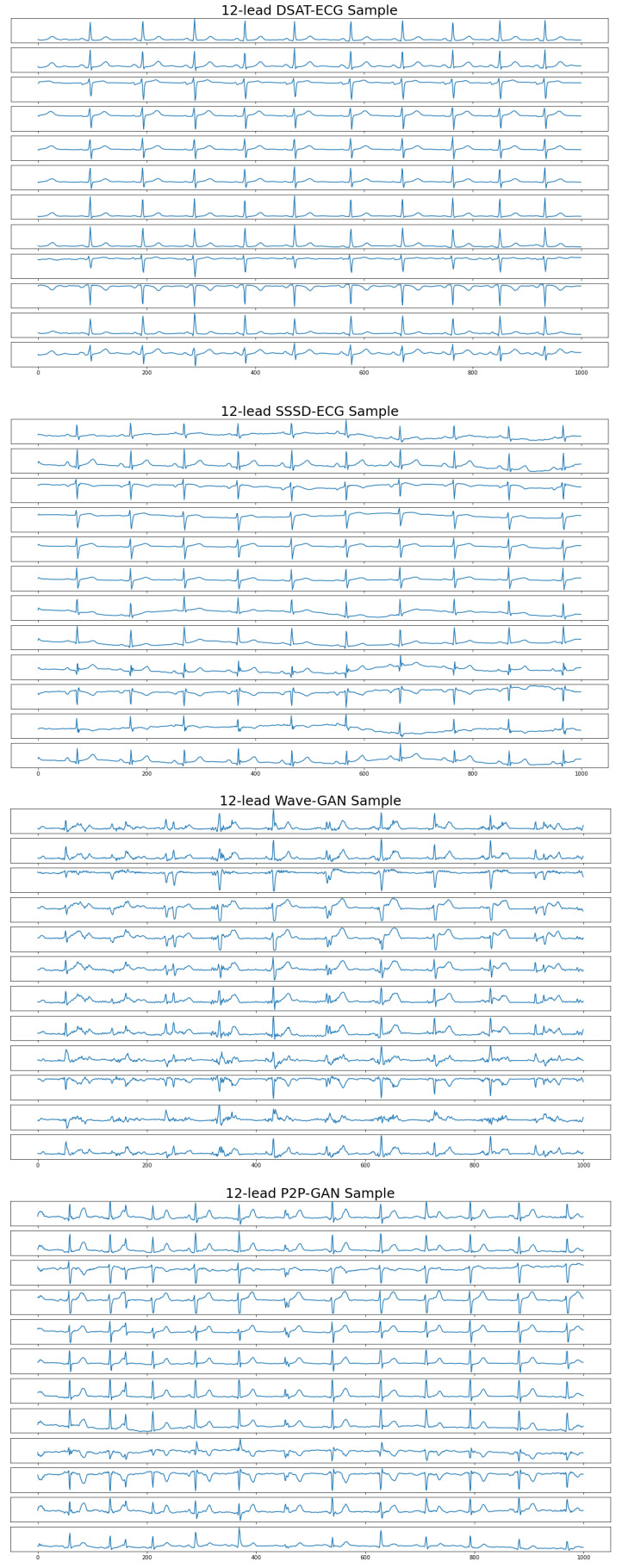
Twelve-lead ECG sample generated by different models.

**Table 1 sensors-23-08328-t001:** Quality evaluation of ECG samples.

Model	Avg DTW (Lower is Better)	Avg MMD (Lower is Better)
WaveGAN	8.057	6.418 × $10^{-3}$
P2PGAN	8.430	6.381 × $10^{-3}$
SSSD-ECG	7.481	4.784 × $10^{-3}$
DSAT-ECG (Ours)	**7.174**	**4.603** × $10^{-3}$

Best results are shown in bold. All the models were evaluated with a batch size of 16.

**Table 2 sensors-23-08328-t002:** Authenticity evaluation of ECG samples.

Test Sample	Accuracy (Higher is Better)	AUROC (Higher is Better)
WaveGAN	89.32	88.34
P2PGAN	89.67	89.58
SSSD-ECG	95.01	93.98
DSAT-ECG (Ours)	**95.84**	**94.56**
Real	* **96.38** *	* **94.67** *

Best results on the synthesized sample are shown in bold and results on the real sample are shown in bold italics. All samples were evaluated in batches of batch size of 16.

## Data Availability

Not applicable.

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
