# Peer review of "ECG Synthesis via Diffusion-Based State Space Augmented Transformer"

_sensors, 2023, doi:10.3390/s23198328_

Round 1

Reviewer 1 Report

The authors present a novel diffusion-based model coupled with a State Space Augmented Transformer for artificially synthesized data generation. Also, they evaluated how similar the performance of a pre-trained classifier is when applied to both generated and real ECG samples.

In conclusion, details clarify why the proposed approach outperforms competitor models SSSD-ECG, WaveGAN, and P2PGAN.

Minor comments: Correct punctuation errors and large letters throughout the text, especially in chapter 4.3.

Author Response

Dear Reviewer,

thank you for your time and effort reviewing our paper. The paper has been revised carefully according to the reviewers' comments. A response letter is attached, too.

Thanks again! Best wishes, Friedhelm

Reviewer 2 Report

In the introduction, you've quickly transitioned from the general concern of CVDs to AI integration with healthcare. It might be more reader-friendly to provide a brief background on how AI techniques are currently being applied in the CVD domain before introducing your specific method. There is a missing period or other punctuation after "Maximum Mean Discrepancy (MMD)" in line 10. In line 3, you mentioned "privacy concerns," and in line 4, you again mention "privacy concerns." Consider rephrasing one of them for variety. The sentence structure in lines 6-7, especially the phrase "This synthesizes 12-lead Electrocardiograms under 12 multi-labeled heart rhythm classes," could be broken down or elaborated further for clarity. It might be helpful to provide more context on why 12-lead ECGs are significant and why having 12 multi-labeled heart rhythm classes is crucial. Any limitation of this study?

"Data-drive" in line 4 seems to be a typo. It should be "Data-driven". In line 11, the phrase "For authenticity evaluation we assessed" would read more smoothly as "To evaluate authenticity, we assessed". You mention "privacy concerns" twice in quick succession in lines 3 and 4. In line 4, "Addressing privacy concerns tied to sensitive health data distribution, we leverage artificially synthesized data generation as a promising solution." The connection between addressing the concern and the solution could be made clearer. Perhaps: "To address privacy concerns related to sensitive health data distribution, we propose leveraging artificially synthesized data generation." Consider varying the phrasing to avoid repetition.

Author Response

(The authors gave the same response as above.)

Reviewer 3 Report

Thank you for submitting the manuscript to SENSORS. The research problem addressed in this paper is highly significant and innovative. The proposed method successfully synthesizes 12-lead electrocardiograms using a diffusion-based model and a state space-augmented transformer, and the quality of the generated samples is evaluated using appropriate metrics. The experimental results demonstrate the excellent performance of the proposed method in terms of sample quality and authenticity.

The paper is relatively perfect, but there are some places that need to pay attention to. It is suggested that the authors further enhance the introduction section by providing a detailed overview of the current status of electrocardiogram analysis and the importance of electrocardiogram synthesis, to provide a comprehensive background and discuss related work.

Author Response

(The authors gave the same response as above.)
